# Hyaluronic Acid-Poly(*N*-acryloyl glycinamide) Copolymers as Sources of Degradable Thermoresponsive Hydrogels for Therapy

**DOI:** 10.3390/gels6040042

**Published:** 2020-11-23

**Authors:** Mahfoud Boustta, Michel Vert

**Affiliations:** Institute for Biomolecules Max Mousseron, UMR CNRS 5247, Faculty of Pharmacy, University of Montpellier-CNRS-ENSCM, 15 Avenue Charles Flahault, BP 14491, CEDEX 5, 34093 Montpellier, France; mahfoud.boustta@umontpellier.fr

**Keywords:** thermoresponsive hydrogel, UCST-type thermal transition, hyaluronic acid, poly(*N-*acryloyl glycinamide), hyaluronidase, enzymatic degradation, sustained drug delivery, graft copolymers

## Abstract

One-pot free-radical polymerization of *N*-acryloyl glycinamide in the presence of hyaluronic acid as transfer-termination agent led to new copolymers in high yields without any chemical activation of hyaluronic acid before. All the copolymers formed thermoresponsive hydrogels of the Upper Critical Solution Temperature-type in aqueous media. Gel properties and the temperature of the reversible gel ↔ sol transition depended on feed composition and copolymer concentration. Comparison with mixtures of hyaluronic acid-poly(*N*-acryloyl glycinamide) failed in showing the expected formation of graft copolymers conclusively because poly(*N-*acryloyl glycinamide) homopolymers are also thermoresponsive. Grafting and formation of comb-like copolymers were proved after degradation of inter-graft hyaluronic acid segments by hyaluronidase. Enzymatic degradation yielded poly(*N-*acryloyl glycinamide) with sugar residues end groups as shown by NMR. In agreement with the radical transfer mechanism, the molar mass of these released poly(*N-*acryloyl glycinamide) grafts depended on the feed composition. The higher the proportion of hyaluronic acid in the feed, the lower the molar mass of poly(*N-*acryloyl glycinamide) grafts was. Whether molar mass can be made low enough to allow kidney filtration remains to be proved in vivo. Last but not least, Prednisolone was used as model drug to show the ability of the new enzymatically degradable hydrogels to sustain progressive delivery for rather long periods of time in vitro.

## 1. Introduction

Polymeric hydrogels are 3D matrices that swell in aqueous media [1,2]. Hydrogels are widely implicated in countless biomedical applications [3,4]. The main domains are drug delivery and tissue engineering [5,6,7]. In hydrogels, water is entrapped in a 3D network of crosslinked hydrophilic macromolecules. In contrast to covalent crosslinking, physical crosslinking can be reversed when the environment is no longer favorable. The disappearance or the formation of inter-macromolecule bindings results in sol ↔ gel reversible phase transition. Modifications that cause such transition are referred to as stimuli [8]. Salt concentration, pH, ionic strength, and temperature changes are stimuli exploited to modify water-polymer interactions and provoke reversible sol ↔ gel transitions. In mammalians, temperature is well defined and normally set at physiological values, i.e., 37 °C in humans. This temperature may serve as reference to take advantage of phase transitions in therapeutic applications. For local injection and drug delivery, the transition temperature must be such that it allows injection of a solution followed by in situ gelation as fast as possible [9]. For this, aqueous solutions of natural or artificial polymer-based systems that exhibit a temperature-dependent phase transition close to body temperature are of interest [10]. Many examples are reported in the literature. The more frequent ones are based on Low Critical Solution Temperature-type phase transition (LCST) [11]. In this case, the thermoresponsive polymer-water system is a solution at low temperature and a more or less condensed separated phase (gel or precipitate) above the phase transition temperature. The archetypes of polymers leading to LSCT-type transitions are A-B-A and B-A-B poly(lactic acid)-poly(oxyethylene)-poly(lactic acid) triblock copolymers and poly(*N-*isopropyl acrylamide (PNIPAM)-based polymers. In these cases, the formation of the condensed phase is due to dehydration of hydrophilic parts of the constitutive macromolecules and the expulsion of water leads to more or less contraction relative to the volume of the solution phase. Another class of thermoresponsive hydrogels is based on the phenomenon of Upper Critical Solution Temperature (UCST) [12]. They are less frequent in the literature of therapeutic applications because both the temperature of the injected solution and that of the in situ sol → gel phase transition have to be slightly above 37 °C. Administration of a solution at much higher temperature may cause cell and tissues injuries prior to gel formation. Once suitable biocompatible administration conditions are tuned, UCST-based hydrogels are attractive because the hydrogen bond-based crosslinking is destabilized without dehydration and loss of hydrophilicity. Therefore, the volume of the solution is preserved as in the case of gelatin, for instance [13].

In a previous publication, we have shown that high molar mass poly(*N-*acryloyl glycinamide) polymers (PNAGAs) can fulfill many of the requirements related to sustained delivery after administration by injection of a warm solution the temperature of which being tuned through polymer molar mass and concentration [14]. Disappearance of a PNAGA gel was observed a few days after intraperitoneal injection in mice in apparent contradiction with the renowned resistance to biodegradation of polyacrylic backbones. The reason for this disappearance remained unexplained although it may have been related to slow hydrolysis of amide functional groups of PNAGA side chain in aqueous medium [15]. If it was the case, the disappearance of the hydrogel was not due to degradation or biodegradation of the polyacrylic main chain but preferably to dissolution of the generated carboxylic polyanion.

In attempt to overcome the absence of main chain degradation that may be a serious drawback in the perspective of applications in human, we decided to investigate the possibility of combining hyaluronic acid (HA), a well-known biodegradable polymer, with the UCST-type PNAGA thermoresponsive polymer (Figure 1), hoping the formation of new thermoresponsive copolymers and maybe bioresorbable if the molar mass of degradation by-products is low enough to allow kidney filtration [16]. Hyaluronic acid is a linear glycosaminoglycan composed of d-glucuronic acid and *N-*acetyl-d-glucosamine linked together through alternating β-1,4 and β-1,3 glycosidic bonds. It is present in human and animal tissues and as a non-Newtonian viscous fluid in joint cavities (synovial fluid) and eyes (vitreous). It is also a constituent of the extracellular matrix (ECM). Hyaluronic acid is a biopolymer easily accessible and degradable in vivo and in vitro by hyaluronidase, a specific enzyme [17]. The degrading action of hyaluronidase on chemically modified HA was shown possible in many instances [18,19,20,21]. In contrast, the gels formed by PNAGA polymers are thermoresponsive but not degradable.

This article reports on the synthesis, characterization and enzymatic degradation of HA-PNAGA comb-like copolymers. A one-pot original synthesis was selected that requires neither activation nor preliminary chemical modification of HA molecules. Various analytical techniques were used to prove the formation of thermoresponsive HA-PNAGA copolymers and the HA main chain degradation by hyaluronidase to release pendent PNAGA segments with rather low molar mass. Only the enzymatic degradation in the presence of hyaluronidase was successful in proving the expected comb-like structure. Last but not least, the ability of HA-PNAGA hydrogels to allow sustained drug delivery was explored using Prednisolone as drug model. Data and findings are discussed in perspective of potential applications of the resulting new thermoresponsive hydrogels in the biomedical area.

## 2. Results and Discussion

### 2.1. Synthesis

It has been known for several decades that free-radical polymerization of *N*-acryloyl glycinamide (NAGA) in water can be controlled using isopropanol as transfer agent of radicals from growing chains to monomer molecules [22]. The method is general and it has been largely exploited in polymer synthesis to initiate new polymer chains and limit molar mass [23]. The process leaves a residue of the transfer agent attached to the beginning of the new macromolecule. This particularity is generally neglected because end groups have little effects on macroscopic properties relative to the rest of macromolecules. In the present work, radical transfer was tentatively applied to add pendent PNAGA segments to HA molecules in one-pot assuming that some of the many alcohol functional groups may act each as on-macromolecule transfer agent with grafting of PNAGA side chains as result, this without any intermediate chemistry (Figure 2).

Feeds composed of different proportions of HA and NAGA were polymerized in water after initiation using sodium persulfate as water-soluble initiator. All the polymeric compounds (assumed HA-PNAGA comb-like copolymers) recovered after purification by dialysis showed reversible UCST-type gel ↔ sol transitions (Table 1). Data show that all the feeds led to polymers in high yield (≥95%) after purification by dialysis. Only very small amounts of unreacted NAGA were eliminated during the purification process. All the polymers retained inside the dialysis tube formed gels provided their concentration in the aqueous medium was above a minimum dependent on the initial composition in HA and NAGA. In general, cooling a hot solution to room temperature has led to a gel within less than two minutes (Table 1).

The observation of UCST-type phase transitions did not prove binding of PNAGA segments to HA molecules or formation of comb-like copolymers since, on the one hand, PNAGA itself is thermoresponsive and, on the other hand, mixtures of PNAGA_121_ and HA exhibited gel ↔ sol transitions too, as exemplified by the comparison between HA_100_-PNAGA 8 and HA_100_ + PNAGA_121_ (Table 1). The corresponding sol → gel transition temperatures were different but this difference could not be considered either as proof of grafting because the PNAGA_121_ that was used as model of the PNAGA grafts present in HA-PNAGA polymeric compounds was not representative of the molar mass of the PNAGA segments expected to be present in the recovered polymers. Other analytical techniques had thus to be involved to prove grafting.

### 2.2. Spectroscopic Analyses

The recovered polymers were first analyzed by FTIR spectroscopy in comparison with HA and PNAGA precursors and a mixture of HA + PNAGA as exemplified in Figure 3. The main bands of HA (1034 cm^−1^) and PNAGA (1547, 1259 and 1034 cm^−1^) were present in both the recovered polymer and the HA + PNAGA mixture. However, there was no exploitable difference between the spectrum of the copolymer and that of the HA + PNAGA mixture with the same composition in agreement with the very small number of HA-PNAGA junctions in the expected comb-like copolymer macromolecules.

Spectroscopic comparison was then made using ^1^H-NMR. For instance, the spectrum of HA_30_-PNAGA 6 exhibited the resonances typical of HA,Na and PNAGA (Figure 4). However, a shoulder peak at 4.05 ppm present in HA_30_-PNAGA 6 was absent in the spectrum of the mixture. The same extra peak was observed for HA_100_-PNAGA 8.

Figure 5 shows that the shoulder peak at 4.05 ppm was dependent on the feed composition and grew parallel to the growth of the complex HA signals between 3.6 and 3.9 ppm. According to radical transfer, more HA in the feed led to more grafting of smaller grafts. Therefore, the shoulder peak could have been considered as reflecting the protons of the HA-PNAGA junctions but overlapping and the lack of reference precluded doing so. ^13^C-NMR was also exploited the same way. All the ^13^C-NMR spectra reflected the presence of the two polymeric components but there was no extra band assignable to HA-PNAGA linkage in this case.

Facing the absence of direct demonstration of the coupling by FTIR and NMR spectroscopy, copolymer formation had to be demonstrated indirectly from clues supplied by other means.

### 2.3. Indirect Demonstration of the Grafting of PNAGA onto HA Macromolecules

#### 2.3.1. Physical Aspect

Figure 6 shows the aspects of gels of HA_100_-PNAGA 5, HA_100_ + PNAGA_121_ with the same gross composition, and PNAGA_121_ alone; the three-system containing the same concentration in PNAGA corresponding to 5% *W*/*V*. Under these conditions, the PNAGA gel was quasi transparent. In contrast, the gelled mixture was turbid while HA_100_-PNAGA 5 was only slightly turbid. The turbidity was assigned to incompatibility between the two polymeric components in the mixture. The low turbidity of HA-PNAGA 5 relative to the turbidity of the mixture suggested a structural difference but could not be considered as proof of copolymer formation.

#### 2.3.2. Molar Mass Determination

Visometry and SEC were used in attempt to estimate the molar masses of HA-PNAGAs relative to parent HAs. However, water and DMSO were the only useable solvents. None led to exploitable data because of the formation of nanosized aggregates as in the case of PNAGA alone [24]. The use of H-bond disrupting salt (LiBr) failed, the aggregates being only partially destabilized.

Maldi-Tof was also considered but no matrix provided consistent information. At least, there is no doubt about the polymeric nature of the recovered compound, HA being already a polymer.

#### 2.3.3. Ionic Interaction with Cationic Poly(l-Lysine)

The formation of polyelectrolyte complexes (PEC) was tentatively exploited to distinguish HA_100_-PNAGA and HA_100_ + PNAGA_121_ of similar compositions. HA is a negatively charged polymer whereas PNAGA is neutral. Fluorescein-labelled poly(l-lysine), (PLLFlu), was used as colored polybase for complexation with HA. The HA_100_ + PNAGA_121_ mixture led to a colored precipitate that decanted immediately in accordance with the behavior of HA alone. In contrast, the HA-PNAGA compound led to a stable turbid suspension. The absence of macroscopic precipitation argued in favor of a copolymer structure and well agreed with the small turbidity observed visually (cf. 2.3.1). Polyanion-polycation electrostatic complexation leads normally to dehydration and precipitation as it was the case for HA mixed with PLLFlu. In contrast, the HA-PNAGA-PLLFlu complex appeared as slightly turbid suspension of dispersed sub-micrometric aggregates as it is usually the case of dispersion of PEC’s in the present of surfactants or of an excess in one of the polyions [25]. In the case of HA-PNAGA-PLLFlu, the stable dispersion of aggregates was considered as a signature of the amphiphilic structure for the HA-PNAGA copolymer in which hydrophilic side-chains prevented macroscopic precipitation of the dehydrated PLL/HA-PNAGA complex. This feature argued strongly in favor of the expected comb-like structure of HA-PNAGAs.

#### 2.3.4. Degradation by Hyaluronidase

Hyaluronidase is an enzyme present in humans. It degrades hyaluronic acid in vivo and in vitro as well. The preservation of this property has been reported for many comb-like copolymers provided the degree of grafting was not too high. Accordingly, hyaluronidase was exploited in attempt to enzymatically degrade the HA main chain of HA-PNAGA copolymers in order to release PNAGA grafts. The principle of the degradation with release of grafts is depicted Figure 7.

The NMR spectrum of fractions retained in the dialysis tube (Figure 8A) was rich in PNAGA in agreement with resonances typical of this polymer shown in Figure 4. In contrast, these resonances were very small for the compound that had diffused through the dialysis membrane. The spectrum was dominated by resonances of the fragments formed during the degradation of HA main chains (Figure 8B). These features proved the formation of comb-like copolymers conclusively. They also gave value to the trends and clues reported in the previous section.

To confirm this conclusion, comparison was made of the concentration-dependence of gel → sol transition temperature for PNAGA-rich by-products issued from the dialysis of HA-PNAGA copolymers having different molar mass and different HA content. The corresponding profiles are presented in Figure 9. The molar mass of HA was not an important factor as shown by the close profiles of HA_30_ and HA_100_-derived PNAGA-rich residues that had similar compositions. In contrast, the proportion of HA was more important. Increasing the HA proportion from 20 to 50% in the parent HA-PNAGA copolymer led to a large shift of the profiles of PNAGA-rich residues toward low concentrations. This finding argued in favor of a decrease of the molar mass of residual PNAGA when the HA proportion increased from 20 to 50% HA in agreement with the trend expected when the concentration in transfer agent increases. The trend was confirmed by comparing the profiles of residual PNAGAs with the profiles specific to PNAGA homopolymers with 121,000 and 41,000 g/mol determined under similar conditions (Figure 9).

Therefore, increasing the HA proportion in the polymerization feed was an efficient means to lower the molar mass of the PNAGAs degradation by-product left after degradation of HA backbone. To further decrease the molar mass of PNAGA grafts, a HA-PNAGA copolymer was synthesized using 2.5 M HA instead of 1 M to polymerize a 50/50 *w/w* mixture of HA and NAGA. After degradation, dialysis and recovery of PNAGA present in the retentate, and determination of the concentration dependence of the gel → sol transition, it was found that the profile correspond to molar mass significantly lower than 41,000 g/mol, as expected. More precise evaluation was not possible due to lack of low molar mass PNAGA reference.

### 2.4. Ability to Sustaining the Release of a Drug

In order to check the ability of the new HA-PNAGA thermoresponsive gels to sustain drug delivery as PNAGA homopolymers do, two different HA-PNAGA gels (HA-PNAGA 2 and 5) were formulated with poorly soluble Prednisolone. Figure 10 shows the in vitro release profiles provided by these formulations.

After a small burst of c.a. 5%, the release profiles were progressive, and went up to complete 100% release as already observed for PNAGA thermoresponsive hydrogels [14]. To exemplify the possibility of modifying the profiles, two different systems were compared. Data in Figure 10 shows that type of gel and copolymer concentration are means to modify and tune the release rate and duration. As well-known from other drug delivery systems, the release characteristics from the new HA-PNAGA thermoresponsive hydrogels will depend also on the physico-chemical characteristics of the loaded drug thus making deeper investigation on a model compound of little interest beyond illustration of sustained release ability in this study.

## 3. Conclusions

This work showed that PNAGA segments of different sizes can be successfully grafted to HA macromolecules of different sizes in one pot using multi-alcohol HA as macromolecular transfer agent. The comb-like copolymer structure of the resulting thermoresponsive copolymers could not be demonstrated from UCST-type sol-gel transition temperature data or from FTIR and NMR spectroscopic analyses. We had to call on clues provided indirectly by different techniques. It is the enzymatic degradation that demonstrated the comb-like structure conclusively and proved the direct coupling of side-chain PNAGA segments to HA molecules without intermediate chemical activation of HA. It was also shown that thermoresponsive HA-PNAGA hydrogels were able to release progressively Prednisolone over rather long periods of time and may thus be a versatile source of degradable and injectable drug delivery systems. Since characteristics and behavior of HA-PNAGA gels can be tuned via composition and concentration, these new degradable hydrogels may find applications in soft and hard tissue surgery and traumatology as well. Bringing a new in vitro-degradable biomaterial or therapeutic device up to clinical applications is a long route that includes fulfilling many more or less specific requirements [26,27]. Accordingly, the next stage after synthesis and characterization will be collecting information on biocompatibility, on in vivo degradation, and on the possibility to excrete PNAGA degradation by-products via kidney filtration, an important criterion to consider these copolymers as bioresorbable.

## 4. Materials and Methods

### 4.1. Materials

#### 4.1.1. Chemicals

30 KDa HA,Na (HA_30_) and 100 KDa HA,Na (HA_100_) were kindly supplied by Novozymes (Copenhagen, Denmark), whereas 1,5-2,2 MDa HA,Na (HA_2000_) was purchased at Acros Organics Fisher Scientific (Ilkirch, France). NAGA monomer was synthesized from acryloyl chloride and glycinamide in water according to a protocol reported previously [14]. Potassium persulfate (K_2_S_2_O_8_), Sodium azide (NaN3), FITC-labelled poly(l-lysine) (5000–30,000 g/mol), Type I-S Hyaluronidase from bovine testes as lyophilized powder (400–1000 units/mg), Prednisolone, and PBS Dulbecco (PBS) were all purchased at Sigma-Aldrich (Saint-Quentin Fallavier, France). The model PNAGA_121_ polymer with MW = 121,000 g/mol was synthesized in the laboratory.

#### 4.1.2. Synthesis of HA-PNAGA Copolymer

Typically, an amount of HA,Na was introduced in a 250 cm^3^ round-bottom flask together with deionized CO_2_-free water, a suitable amount of NAGA, and 20 mg of K_2_S_2_O_8_. After slow dissolution, the resulting viscous solution was degassed by gentle bubbling of nitrogen for one hour. The flask was closed up and placed in a water bath at 60 °C for 24 h. The gelled content of the flask was heated up to becoming a solution which was introduced in a dialysis bag (3500 Da cut-off). The content of the closed bag was dialyzing against deionized water. The receiving outer medium was changed several times until no further chemical cross the membrane. The bag was opened and the content freeze-dried. The collected white powder was weighed and yield was calculated relative to the weight of HA,Na + NAGA present in the feed (see Table 1 with details for the different synthesized polymer).

### 4.2. Methods

#### 4.2.1. Spectral Characterizations

ATR-FTIR spectra were recorded using a Spectrum 100 FTIR spectrometer Perkin-Elmer (Villebon-sur-Yvette, France). Polymers were ground and the powder was placed on the ATR plate. ^1^H-NMR and ^13^C-NMR spectra were recorded using a 400 MHz Bruker Avance III HD spectrometer (Wissembourg, France) with temperature set at 60 °C to avoid gelation. Typically, approximately 25 mg of compound was solubilized in 0.6 cm^3^ of deuterated water.

#### 4.2.2. Thermal Characterizations

The gel → sol transition temperature of dried dialyzed polymers was determined according to the reverse test tube method. Typically, a suitable amount of powder was introduced in a 4 cm^3^ glass tube together with deionized water or saline to make 1 cm^3^ of mixture that was placed in a water bath at 80 °C up to total dissolution generally observed after approximately 2 min. The solution was allowed to gel at room temperature. The gel was turned back to solution by heating at 80 °C for 5 min and cooled down to 0 °C at 40 °C/min, left at 0 °C for 30 min, and then heated slowly up to 80 °C at 4 °C/min. A first transition temperature was determined when the gel started to flow along the tube wall. This cycle was repeated 3 times to collect four gel → sol transition temperatures successively. Data were averaged from the last two data. In the case of sol → gel transition, ramp-down temperatures were systematically lower but did not differ by more than 3–4 °C from ramp-up gel → sol ones.

#### 4.2.3. Size Exclusion Chromatography (SEC)

SEC analyzes were carried out using DMSO as mobile phase at a flow rate of 1 cm^3^/min and a Waters 510 liquid chromatography pump equipped with a PolarGel column (Agilent Technologies, Montpellier, France) in series with a Waters 410 differential refractometer. Molecular weights were calculated using Empower Chromatography Software Waters and compared to Pullulan standards.

#### 4.2.4. Enzymatic Degradation

100 mg of HA_100_-PNAGA copolymer 8, or of model 50/50 *w/w* HA_100_ + PNAGA_121_ mixture, was mixed with 2 cm^3^ of isotonic PBS phosphate buffer. 0.03% (*w/v*) of sodium azide (NaN_3_) was added to prevent the development of bacteria. The mixture was heated up to 80 °C and kept at this temperature for 5 min before cooling to form a gel. In parallel, 30 mg of hyaluronidase enzyme was dissolved in 4 cm^3^ of Dulbecco^®^ buffered medium (pH = 7.4 saline solution without calcium or magnesium) also containing 0.03% (*w/v*) of NaN3. 0.84 cm^3^ of this solution was added on top of the gel together with 2 cm^3^ of PBS buffer. The mixture was allowed to react at 37 °C. in a Heidolph^®^ 1000 type incubator (Heidolph Instruments GmbH & CO. KG Schwabach, Germany), until complete digestion. The solution was then heated up to 80/90 °C to inhibit the enzyme. After cooling, the solution was filtered and dialyzed (cut-off 3500 Dalton) against water to remove low molecular weight products. The fraction retained in the dialysis tubing was dried and analyzed by ^1^H-NMR. Comparison was made with the NMR spectrum of the fraction that crossed the dialysis tubing. The same protocol was applied to HA_30_-PNAGA and to the corresponding model mixture using HA_30_ instead of HA_100_.

#### 4.2.5. Formulation of Prednisolone Hydrogels

Typically, an amount of dried gel-forming polymer was introduced in a 4 cm^3^ flat-bottom test tube. 1 cm^3^ of PBS buffer was added and the mixture was heated up to dissolution using a water-bath. A suitable sample of Prednisolone was added to the hot solution and the tube containing the resulting suspension was immediately dipped in an ice-bath for immediate gelation to keep the tiny prednisolone particles dispersed.

#### 4.2.6. Prednisolone Sustained Release Profiles

A calibration curve was constructed using standard solutions made by suitable dilutions of a stock solution comprising 6.8 mg Prednisolone in 50 cm^3^ PBS in order to cover the 2.6–54.4 µg/cm^3^ range. In this range, the variation of the optical density at 247 nm was linear. 1 cm diameter glass tube containing 1 cm^3^ of gel formulation of interest was introduced in a 0.5 dm^3^ plastic bottle containing 300 cm3 of receiving medium. The bottle was placed on the oscillating plateau of a Heidolph 1000 incubator thermostated at 37 °C. The tube was allowed to roll back and forth gently during the whole release period. At each selected time point, 1 cm^3^ aliquot was withdrawn that was not replaced by fresh medium given the large volume of receiving medium. Each data point was averaged from duplicated experiments with uncertainties depending on the concentration in the aliquot as indicated by error bars.

## Figures and Tables

**Figure 1 gels-06-00042-f001:**
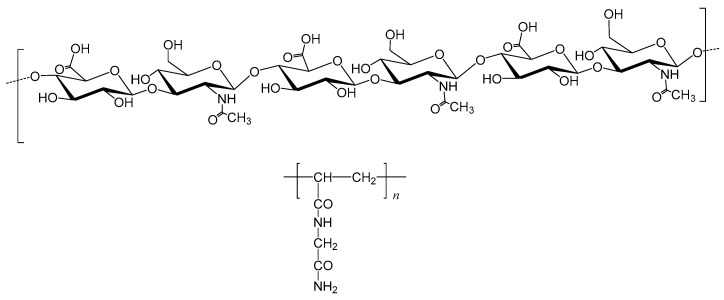
Chemical formula of hyaluronic acid (**top**) and of PNAGA (**bottom**).

**Figure 2 gels-06-00042-f002:**
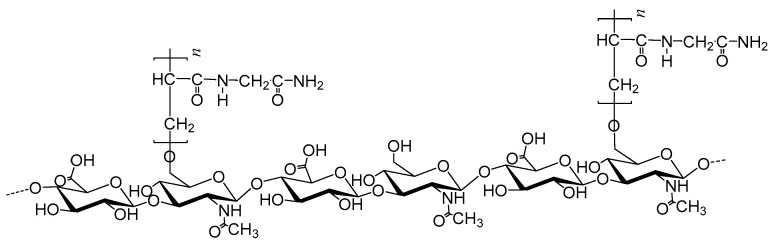
Schematic representation of the formula of a HA-PNAGA comb-like copolymer assuming partial binding at OH sites of HA acting as both transfer agent and macromolecular end-residue.

**Figure 3 gels-06-00042-f003:**
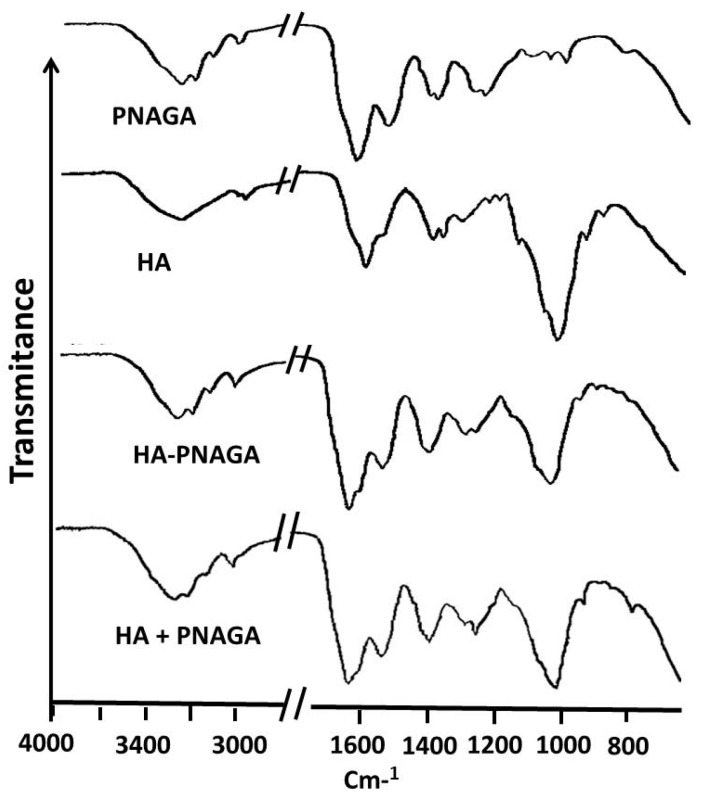
FTIR spectrum of the polymer synthesized from 50% of HA_100_ in the polymer feed (HA_100_-PNAGA 8 in Table 1) compared with the spectra of HA_100_ and PNAGA_121_ precursors and of a 50/50 *w/w* mixture of HA + PNAGA_121._

**Figure 4 gels-06-00042-f004:**
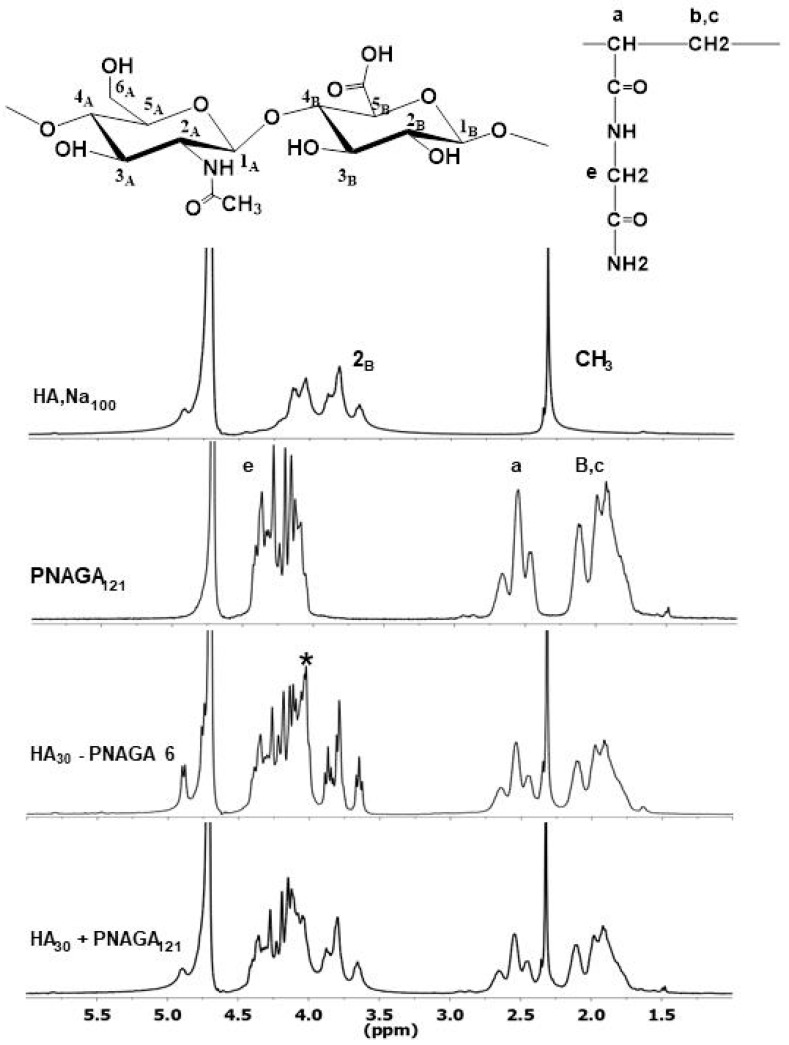
Comparison between ^1^HNMR spectra of parent HA_100_,Na and PNAGA homopolymers, of HA_30_-PNAGA 6 and of mixture of HA_30_ + model PNAGA with the same composition. (* extra peak in the expected copolymer spectrum relative to the mixture; a–c,e: assignments of resonances shown in the spectra correspond to those provided in the formula).

**Figure 5 gels-06-00042-f005:**
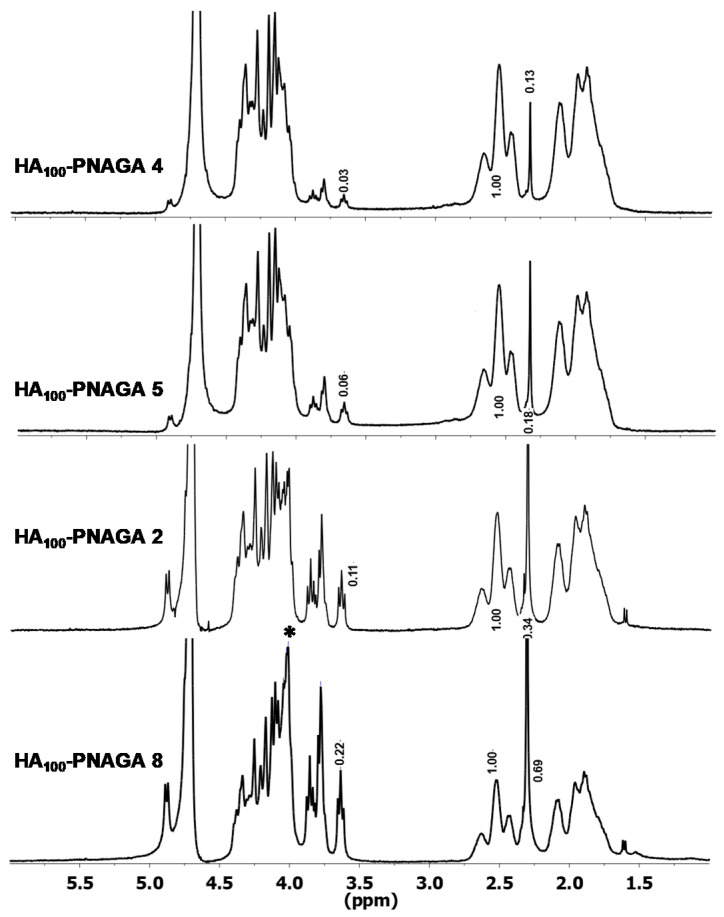
Comparison of ^1^HNMR spectra of HA-PNAGA 4, 5, 2, and 8 to show the increase of the 4.05 peak with the content in HA when the spectra are normalized to the weight of the PNAGA CH triplet set at 1.0 relative weight. (* composition-dependent extra peak in the expected copolymer spectrum relative to the mixture).

**Figure 6 gels-06-00042-f006:**
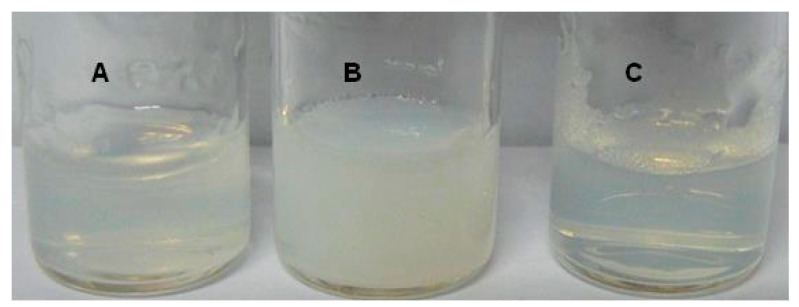
Comparison of the aspects of gels composed of HA_100_-PNAGA 5 (**A**), mixture of HA_100_ + PNAGA_121_ with the same gross composition (**B**), and PNAGA_121_ alone (**C**), all at the same 5% *w*/*v* concentration in PNAGA.

**Figure 7 gels-06-00042-f007:**
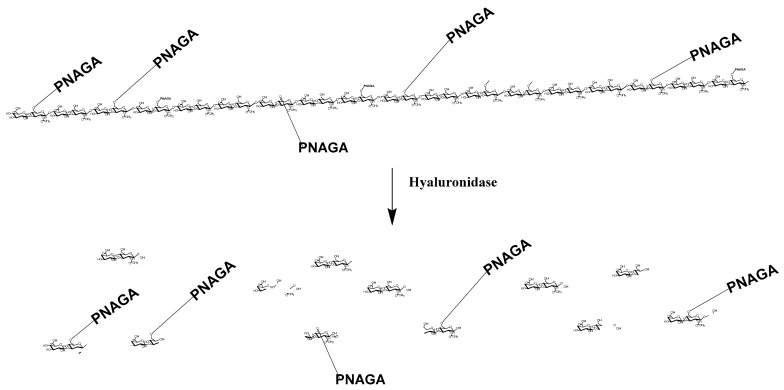
Schematic representation of the enzymatic degradation of a HA-PNAGA in the case of a comb-like structure. The degradation liberates the PNAGA segments with a few sugar fragments of the initial HA chain at the end as shown by NMR Figure 8B. (sugar residues in main chain and residual fragments are represented condensed).

**Figure 8 gels-06-00042-f008:**
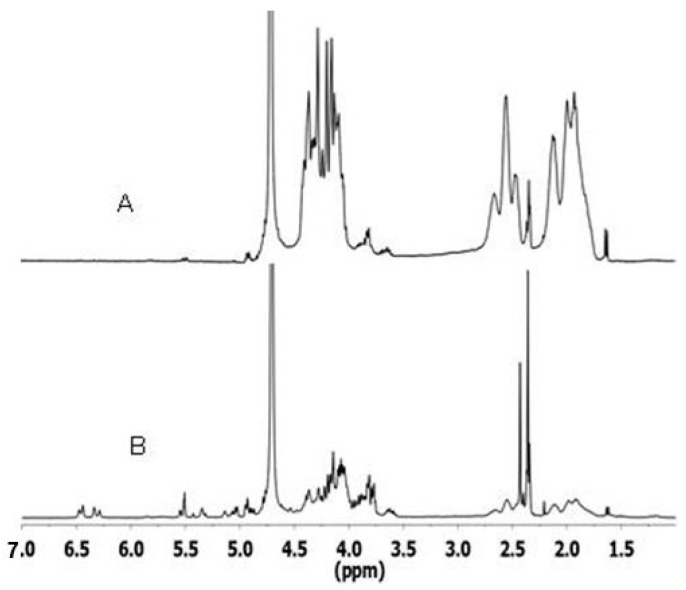
^1^H-NMR spectra of dialyzed residues collected after enzymatic degradation of HA-PNAGA 8 by hyaluronidase: (**A**) retained in the tube, and (**B**) grouped fractions outside the tube.

**Figure 9 gels-06-00042-f009:**
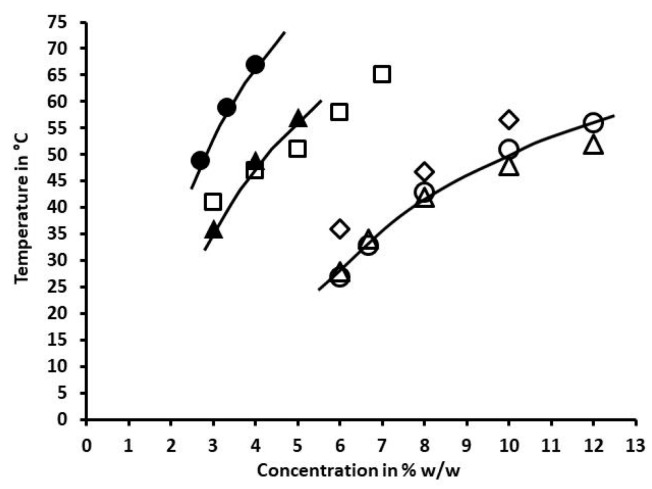
Dependences of the gel → sol transition temperature of gels made from PNAGAs by-products collected from dialysis retentates after enzymatic degradation of different HA-PNAGA in comparison with two PNAGAs of known molar mass (121,000 (◊), and 41,000 (□) g/mol): (∆)HA_30_-PNAGA 6, and (○) HA_100_-PNAGA 8 (50% *w/w* HA in the feed); (▲) HA_30_-PNAGA 1, and (●) HA_100_-PNAGA 2 (20% *w/w* HA I the feed).

**Figure 10 gels-06-00042-f010:**
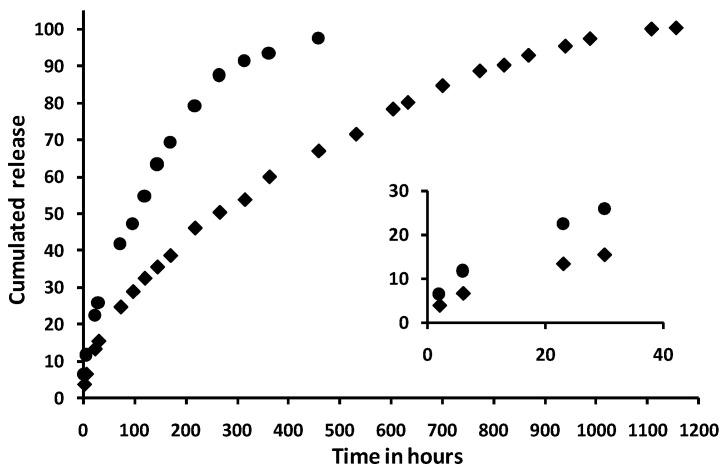
Slow release profiles obtained at 37 °C from 1 cm^3^ of two different formulations of HA-PNAGA gels with the following compositions: (♦) 4% *w*/*v* HA_100_-PNAGA 2 + 20 mg prednisolone (0.055 mM), and (●) 3% *w*/*v* HA_100_-PNAGA 5 + 15 mg Prednisolone (0.042 mM). (Each data point is averaged from duplicates and the difference between the duplicates was well inside the size of the symbols for each profile).

**Table 1 gels-06-00042-t001:** Different HA + NAGA feeds submitted to free-radical polymerization at 60 °C and corresponding gel-sol transition temperatures of the recovered polymeric compounds after purification by dialysis.

Polymers	HA,Na ^a^ g	NAGA ^a^ g	HA/ HA + NAGA ^b^ *W*/*W* %	HA/ HA + NAGA ^c^ g *W*/*W* % (Mol_disac_%)	Yield % ± 2	Gel → Sol Transition Temperature ^d^ °C
HA_30_ − PNAGA 1	0.3	1.2	20		97	55–56
HA_100_ − PNAGA 2	0.375	1.5	20	23 (7.4)	96	71–72
HA_100_ − PNAGA 3	0.065	2.0	3.15		96	66–68
HA_100_ − PNAGA 4	0.25	2.5	9.18	9.2 (2.93)	95	83–84
HA_100_ − PNAGA 5	0.4	2.5	13.8	17.7 (5.7)	95	85–87
HA_30_ − PNAGA 6	0.4	0.4	50	54.4 (17.4)	97	43–45 ^e^
HA_100_ − PNAGA 7	0.4	0.4	50		98	41–42 ^e^
HA_100_ − PNAGA 8	0.4	0.4	50	56.4 (18)	94	42–43 ^e^
HA_2000_ − PNAGA 9	0.4	2.5	13.8		96	77–79
HA_2000_ − PNAGA 10	0.2	2.5	9.18	9.2 (2.9)	95	73–75
HA_30_ + PNAGA_121_	0.01	0.01	50	50 (16 ± 1)	-	-
HA_100_ + PNAGA_121_	0.01	0.01	50	53 (17 ± 1)	-	58–59 ^f^

^a^ In the polymerization mixture; ^b^ Relative to the amounts of HA and NAGA in the polymerization feed; ^c^ From ^1^H-NMR after purification by dialysis against water; ^d^ 5% *w/w* in water; ^e^ There was no gel formed at 5% *w/w* in water. Data were collected for 12% *w/w*; ^f^ This mixture made at 12% *w/w* to obtain a gel that could be compare with ^e^ HA was viscous and the gel-sol transition observed was assignable to the PNAGA which was then at 6% *w/w*.

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
