# Peer review of "Hyaluronic Acid-Poly(N-acryloyl glycinamide) Copolymers as Sources of Degradable Thermoresponsive Hydrogels for Therapy"

_gels, 2020, doi:10.3390/gels6040042_

Round 1
Reviewer 1 Report
In this study, the authors synthesized poly(N-acryloyl glycinamide) grafted with hyaluronic acid. In addition, the authors characterized the copolymers and also the gel formed by poly(N-acryloyl glycinamide) grafted with hyaluronic acid. The work is interesting. However, at the current stage, the work may not be accepted for publication. Revision is recommended. Please see the comments below.
- In addition to NMR characterization, the authors should provide FTIR spectrum of the co-polymers to indicate two component hydrogel or copolymer.
- What are the PDI of the polymer and also please show the molecular weight of all the co-polymers in the units of g/mol
- The drug release profiles need standard deviations. In addition, at what the temperature was the drug release experiment conducted?
- Is the drug release profile different at different temperatures since the co-polymer is temperature responsive?
- Why did the authors particularly pick 4% W/V HA 100 -PNAGA 2 and 3% W/V HA 100 -PNAGA 5 for drug release study? It would be interesting to check the various co-polymers with different PNAGA and HA ratios (percentages) to see if how these factors affect the release kinetics. In addition, the drug amount should be the same for comparison.
- For drug delivery application, the authors should provide in vitro cytotoxicity data of the hydrogel.
Author Response
Point-by-point reply to reviewers’ comments and requests
We thank the reviewers very much for their reports and the comments and requests therein. We submitted our work to Gels because the main topics addressed in our paper are related to original synthesis, characterization and demonstration of structure of new gels as recognized by the reviewers. Because these gels are of potential interest for therapeutic applications, especially sustained and maybe controlled drug release (for the difference between these two terms often used as equivalent see "Terminology for biorelated polymers and applications (IUPAC Recommendations 2012)," Pure and Applied Chemistry, vol. 84, pp. 377-408, 2012), we added an example to illustrate that these gels are able to sustain the delivery of a model drug indicated as a main target in the introduction. To be of interest, deeper investigations have to be performed on a real drug which a specific disease to treated since each drug /delivery system is a special case and related work are generally published in specific journal like Journal of Controlled Release, Drug targeting, and the same. Therefore, our point-by-point reply was constructed accordingly.
Reviewer 1
In this study, the authors synthesized poly(N-acryloyl glycinamide) grafted with hyaluronic acid. In addition, the authors characterized the copolymers and also the gel formed by poly(N-acryloyl glycinamide) grafted with hyaluronic acid. The work is interesting. However, at the current stage, the work may not be accepted for publication. Revision is recommended. Please see the comments below.
- In addition to NMR characterization, the authors should provide FTIR spectrum of the co-polymers to indicate two component hydrogel or copolymer.
In response to reviewer suggestion, a Figure (3) was added to provide FTIR spectra and to emphasize the initial statement that there was no difference between an expected copolymer and the model mixture of components.
- What are the PDI of the polymer and also please show the molecular weight of all the co-polymers in the units of g/mol
We would have like to provide these information’s but, as indicated in the text, Size Exclusion Chromatography was inappropriate because of the presence of aggregates even when measurements were performed at high temperature in the presence of H-bond breaking salts like LiBr or KSCN. The use of Maldi-Tof was also attempted without success because of a lack of suitable matrix. Starting from HMW HA, we are sure the synthesized compounds are polymers. The text was modified accordingly with indication of viscometry and Maldi failures in addition.
As for the general use of g/mol, we used this unit for PNAGA polymers and Dalton for the HA biopolymer as it is commonly done. Dalton is used for water –soluble biomacromolecules like HA. So we prefer to respect the common units for each types of component. By the way Dalton is basically not g/mol from a physical-chemistry point of view.
- The drug release profiles need standard deviations. In addition, at what the temperature was the drug release experiment conducted?
Because the number of replicate was small, we did not indicate SD. Instead we indicated mean values of duplicates or triplicates with maximal uncertainties. This is now indicated in the caption of Figure 9.
The temperature at which drug release profiles were obtained was 37°C as indicated in the experimental section. The temperature was added in the caption.
- Is the drug release profile different at different temperatures since the co-polymer is temperature responsive?
Thermoresponsive or not, release from a gel or from any drug delivery matrix does depend on temperature. Anyhow, such information requests deeper investigation and its interest is relevant to investigation of drug release at different temperatures as part of academic investigations. In our case, the release is expected at body temperature and the provided illustrating profiles were collected at 37°C as indicated in the experimental section and now in the caption.
- Why did the authors particularly pick 4% W/V HA 100 -PNAGA 2 and 3% W/V HA 100 -PNAGA 5 for drug release study? It would be interesting to check the various co-polymers with different PNAGA and HA ratios (percentages) to see if how these factors affect the release kinetics. In addition, the drug amount should be the same for comparison.
The reviewer was right. This kind of deeper investigations is relevant to a specific drug/DDS case. We intend to follow the advice in the future when we will have to deal with specific drug and disease. Anyhow, a sentence was added to justify the selected experimental conditions as part of exemplification.
- For drug delivery application, the authors should provide in vitro cytotoxicity data of the hydrogel
Correct and rather general in work dealing with drug delivery. However, when application in human is the issue, cytotoxicity in vitro is of little interest relative to toxicity in vivo. Anyhow both require specific deep investigations once the copolymers are well identified. Of course, we intend to do this kind of necessary investigation in the future. Preliminary tests at the level of a PNAGA precursor have already been engaged.
Reviewer 2 Report
The paper entitled “Hyaluronic acid-poly(N-acryloyl glycinamide) Copolymers as Sources of Degradable Thermoresponsive Hydrogels for Therapy” by authors Mahfoud Boustta and Michel Vert discuss UCST thermo-responsive copolymers made by one-pot free-radical polymerization of N-acryloyl glycinamide (PNAGA) in the presence of hyaluronic acid (HA). They investigate feed composition and copolymer concentration effects on the gel-sol transition. Formation of graft copolymers with comb-like structure was shown upon degradation (by enzyme hyaluronidase) of inter-graft hyaluronic acid segments using NMR technique. They also used prednisolone, as a model drug, to show the ability of the enzymatically degradable hydrogel (HA-PNAGA) to release (in vitro) the drug for a prolonged period of time (hundreds of hours). The findings presented suggest potential biomedical applications of these new UCST hydrogels.
The paper is well written, scientifically sound, and the results well presented. I especially appreciate the creative approach to the identification of the grafted comb-like structure of synthesized polymers. Overall, I have only minor issues, which if adequately addressed the paper should be published in Gels:
1) Ln.109: Although it is evident that acronym NAGA stands for the monomer, NAGA is here used for the first time and should be then defined.
2) Ln.179: “… 5W/W % in water. Data were collected for 12% WW” should be “… 5% W/W in water. Data were collected for 12% W/W”.
3) Figure 5: The letters denoting the vials should be made legible. Also, the temperature at which the photographs were taken should be noted.
4) Ln.271: “Figure 3” should be “Figure 8”.
5) Ln.301: “Figure 8” should be “Figure 9”.
6) Figure 9, caption: Please specify the temperature at which the experiments were performed (it is essential information).
7) Ln.310: Authors claim: “After a small burst of c.a. 5 %, the release profiles became practically linear…”. What I see in the data (Figure 9) is exponential release profile with saturation. Can authors either explain what they mean by ‘linear’ or change the claim?
Author Response
The paper entitled “Hyaluronic acid-poly(N-acryloyl glycinamide) Copolymers as Sources of Degradable Thermoresponsive Hydrogels for Therapy” by authors Mahfoud Boustta and Michel Vert discuss UCST thermo-responsive copolymers made by one-pot free-radical polymerization of N-acryloyl glycinamide (PNAGA) in the presence of hyaluronic acid (HA). They investigate feed composition and copolymer concentration effects on the gel-sol transition. Formation of graft copolymers with comb-like structure was shown upon degradation (by enzyme hyaluronidase) of inter-graft hyaluronic acid segments using NMR technique. They also used prednisolone, as a model drug, to show the ability of the enzymatically degradable hydrogel (HA-PNAGA) to release (in vitro) the drug for a prolonged period of time (hundreds of hours). The findings presented suggest potential biomedical applications of these new UCST hydrogels.
The paper is well written, scientifically sound, and the results well presented. I especially appreciate the creative approach to the identification of the grafted comb-like structure of synthesized polymers.
Thank you very much for this comment
Overall, I have only minor issues, which if adequately addressed the paper should be published in Gels:
- 109: Although it is evident that acronym NAGA stands for the monomer, NAGA is here used for the first time and should be then defined.
Done
- 179: “… 5W/W % in water. Data were collected for 12% WW” should be “… 5% W/W in water. Data were collected for 12% W/W”.
Done
Figure 5: The letters denoting the vials should be made legible. Also, the temperature at which the photographs were taken should be noted.
Thank you for the detection. Done
- 271: “Figure 3” should be “Figure 8”.
Thank you for the detection. Done
- 301: “Figure 8” should be “Figure 9”.
Thank you for the detection. Done
- Figure 9, caption: Please specify the temperature at which the experiments were performed (it is essential information).
Correct. The temperature was 37°C since it is the temperature at which such a system is supposed to be exploited. The tempérture is now indicated in the caption.
- 310: Authors claim: “After a small burst of c.a. 5 %, the release profiles became practically linear…”. What I see in the data (Figure 9) is exponential release profile with saturation. Can authors either explain what they mean by ‘linear’ or change the claim?
The reviewer is correct; the release profile is not strictly linear as it would be if the model compound was very poorly soluble. The text was changed as suggested, the release mechanism being far aside the main goal of this work.
Round 2
Reviewer 1 Report
Thanks for the authors' response. The authors addressed some of my comments by providing new data. For the suggested experiments in my comments, according to the author's response, the authors are planning to conduct them in the future. Thus, Please briefly describe them (one or two sentences) in the conclusion part as future plans.
Author Response
I am not sure regarding the request of the reviewer. We thought the last sentence in the conclusion that says:
"The next stage will be collecting information on biocompatibility, on in vivo degradation, and on the possibility to excrete PNAGA degradation by-products via kidney filtration, an important criterion to consider these copolymers as bioresorbable."
was indicative enough. However, to follow the reviewer and justify the content of the intended future stragegy, the following has been added before this sentence with two references:
"Bringing a new in-vitro degradable biomaterial or therapeutic device up to clinical applications is a long route that includes fulfilling many requirements [26-27]. Accordingly, the next stage after synthesis and characterization will be .... "